# From Pixels to Words – Towards Native Vision-Language Primitives at Scale

**Haiwen Diao**[1,2*] **Mingxuan Li**[2,3] **Silei Wu**[2] **Linjun Dai**[2]
**Xiaohua Wang**[3] **Hanming Deng**[2] **Lewei Lu**[2] **Dahua Lin**[2] **Ziwei Liu**[1†]

[1]S-Lab, Nanyang Technological University  [2]SenseTime Research  [3]Xi'an Jiaotong University

🖼 **Website:** https://github.com/EvolvingLMMs-Lab/NEO

## ABSTRACT

The edifice of native Vision-Language Models (VLMs) has emerged as a rising contender to typical modular VLMs, shaped by evolving model architectures and training paradigms. Yet, two lingering clouds cast shadows over its widespread exploration and promotion: (-) What fundamental constraints set native VLMs apart from modular ones, and to what extent can these barriers be overcome? (-) How to make research in native VLMs more accessible and democratized, thereby accelerating progress in the field. In this paper, we clarify these challenges and outline guiding principles for constructing native VLMs. Specifically, one native VLM primitive should: (i) effectively align pixel and word representations within a shared semantic space; (ii) seamlessly integrate the strengths of formerly separate vision and language modules; (iii) inherently embody various cross-modal properties that support unified vision-language encoding, aligning, and reasoning. Hence, we launch **NEO**, a novel family of native VLMs built from first principles, greatly narrowing the gap with top-tier modular counterparts across diverse real-world scenarios. With 390M image-text examples, **NEO** efficiently develops visual perception from scratch while mitigating vision-language conflicts inside a dense and monolithic model crafted from our elaborate primitives. We position **NEO** as a cornerstone for scalable and powerful native VLM development, paired with a rich set of reusable components that foster a cost-effective and extensible ecosystem.

## 1 INTRODUCTION

Recently, Vision-Language Models (VLMs) (Bai et al., 2025; Zhu et al., 2025; Wang et al., 2025b; xAI, 2025; Anthropic, 2025; DeepMind, 2025; Hurst et al., 2024; OpenAI, 2025) have emerged as a major breakthrough, extending the strong linguistic capabilities of Large Language Models (LLMs) to multimodal understanding. They typically follow a modular design that integrates a pre-trained Visual Encoder (VE) (Radford et al., 2021; Chen et al., 2024f; Fang et al., 2023; Tschannen et al., 2025), a Projector (Alayrac et al., 2022; Liu et al., 2024a; Dai et al., 2024), and an LLM (Touvron et al., 2023; Yang et al., 2025; DeepSeek-AI et al., 2025). Through multi-stage post-training at scale, they incrementally overcome limitations in image resolution, aspect ratio, and visual encoding flexibility. Yet, modular designs still contend with strong inductive biases in pre-trained visual semantics, complex infrastructure, and scaling laws needed to harmonize their components.

Against this backdrop, native VLMs have arisen as a new avenue of exploration, with Fuyu (Bavishi et al., 2023) and EVE (Diao et al., 2024) pioneering a promising route towards monolithic VLMs. Subsequent efforts seek to learn vision perception from scratch and mitigate vision-language conflicts via visual encoder distillation (Diao et al., 2024; Li et al., 2025b; Wang et al., 2025a; Li et al., 2025a), mixed training data (Lei et al., 2025; Li et al., 2025a), and modality-specific decomposition (Diao et al., 2025; Luo et al., 2024; 2025; Li et al., 2025a). Nonetheless, constructing visual representations via mapping functions inside pre-trained LLMs often hinders efficiency (Chen et al., 2024d; Luo et al., 2024), destabilizes optimization (Team, 2024; Wang et al., 2024b), and disrupts original linguistic knowledge (Diao et al., 2024; Chen et al., 2024d), even under decoupled designs or large budgets (Beyer et al., 2024). Besides, HoVLE (Tao et al., 2025) and HaploVL (Yan et al., 2025)

---

*Work was done during Haiwen's remote collaboration with SenseTime Research. †Corresponding author.

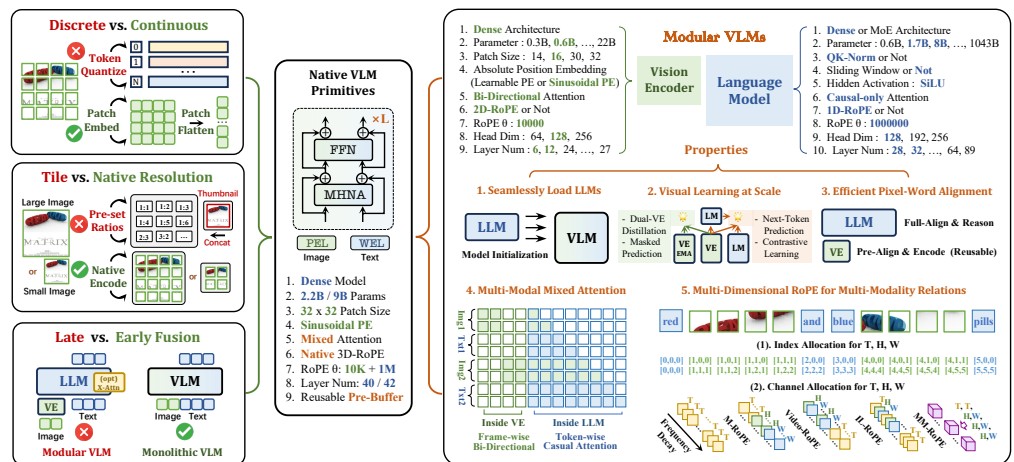

Figure 1: Overview of our native vision-language frameworks, which project arbitrary-resolution images into a continuous latent space, integrating the virtues of modular VLM architectures and enabling efficient vision-language encoding, alignment, and interaction in an early-fusion manner.

address this by first mapping vision-language inputs into a shared space. Yet, their modality-sharing modules, whether derived from LLM or VE layers, neglect intrinsic discrepancy in encoding and interaction across modalities, compromising VLM's capacity to unify visual-linguistic properties.

Figure 1 outlines a central question: *What properties must native VLMs possess to compete with modular ones?* Modular VLMs decouple vision encoders from language models, allowing each to exploit modality-specific characteristics, *e.g.*, bi-directional versus causal attention, distinct positional embeddings, and varied network configurations. This separation accelerates the development of visual and linguistic competencies and permits flexible combinations of individual components. However, it fragments the training procedure, increases alignment costs, and leaves the intermodal balance unresolved. Motivated by these analyses, we formulate the following strategies accordingly:

**(1) Native VLM Primitive.** Native VLMs should embody a unified vision–language primitive that simultaneously integrates encoding, alignment, and reasoning across modalities in one single module. Its design should encompass three principles: (i) a Flexible Position Encoding scheme that generalizes effectively to dynamic spatial structures; (ii) a Multi-Head Native Attention (MHNA) that jointly processes visual–textual connectivity; (iii) Native Rotary Position Embedding (Native-RoPE) that preserves compatibility with pretrained LLM while absorbing VE's interaction patterns. Guided by these tenets, we evolve the LLM blocks into native VLM primitives with brand-new RoPE designs and modality-aware interaction patterns, thereby capturing multi-dimensional relationships for fine-grained and comprehensive correspondence from an intrinsically multimodal perspective.

**(2) Pre-Buffer and Post-LLM.** The next crucial issue is to efficiently scale visual training while securing consistent pixel-word alignment. Here, we partition the monolithic backbone into pre-Buffer and post-LLM layers during pre-training, each rooted in identical native primitive architectures. This transient stage enables pretrained LLMs to steer visual learning and establish coherent relevance with later stages. As mid-training and supervised fine-tuning advance, the partition dissolves, yielding a unified architecture that autonomously allocates the VLM's capacities to their respective functions. This end-to-end training reduces semantic biases of separate pretraining and large overheads of post-stage alignment, effectively bridging native and modular VLMs. Crucially, pre-Buffer persists as a reusable pretrained asset, facilitating sustainable resources for native VLM development.

We launch **NEO**, an innovative native VLM that reimagines multi-modal integration from first principles. Unlike typical modular designs, **NEO** rests on unified primitives that natively encode, align, and reason across modalities, forming coherent pixel–word correspondences from the outset. Through streamlined end-to-end training on 390M image–text samples, **NEO** acquires strong visual perception and approaches leading modular VLMs of comparable scale across diverse benchmarks. Beyond these results, **NEO** offers reusable components that simplify subsequent development and reduce barriers to promoting native exploration. This reveals that next-generation multimodal systems could also originate from architectures that are native, unified, and intrinsically multimodal.

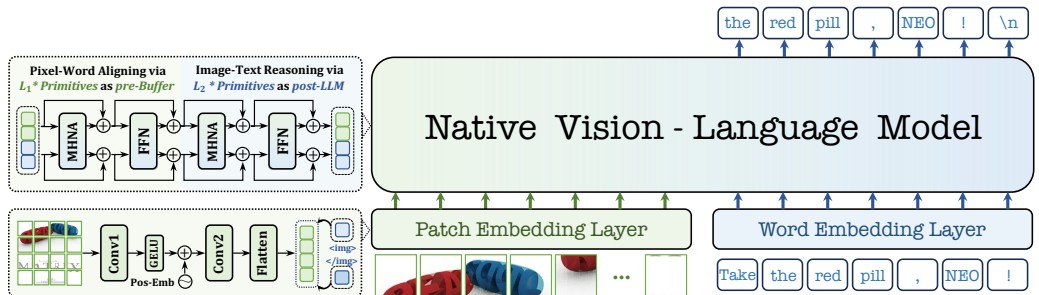

Figure 2: Overview of our proposed NEO architecture. We begin with lightweight patch and word embedding layers that encode images and text into token sequences, which are subsequently processed by a monolithic decoder-only architecture. The pre-Buffer and post-LLM components, each stacked with multiple native primitives, facilitate efficient and precise pixel–word alignment and reasoning.

## 2 RELATED WORKS

### 2.1 MODULAR VISION-LANGUAGE MODELS

Current Vision–Language Models (VLMs) have converged on a modular paradigm, where a pretrained Vision Encoder (VE) is paired with a Large Language Model (LLM) via lightweight adapters, *e.g.*, projection layers (Li et al., 2024a;b) or cross-attention mechanisms (Alayrac et al., 2022; Dai et al., 2024). This encoder-based architecture underlies various popular and leading vision-language systems, including InternVL (Zhu et al., 2025; Wang et al., 2025b), Qwen-VL (Wang et al., 2024a; Bai et al., 2025), Seed-VL (Guo et al., 2025), GLM-V (Hong et al., 2025), and Grok (xAI, 2024; 2025). By harnessing the complementary strengths of vision and language components, modular architectures, adhering to the "ViT-MLP-LLM" pipeline, achieve unprecedented performance across diverse multimodal benchmarks and have emerged as the dominant design principle in the field.

Despite empirical successes, modular VLMs remain constrained by multi-stage training and heterogeneous structures. Extensive post-training interventions are often required to mitigate rigid inductive biases in pretrained VEs (Wang et al., 2024a), which limit resolution flexibility, erode fine-grained details, and blunt sensitivity to features across scales. Besides, pretraining semantic biases and capacity trade-offs between VEs and LLMs collectively impede design simplicity, deployment efficiency, and seamless integration of vision and language, underscoring the urgent need for a monolithic backbone.

### 2.2 NATIVE VISION-LANGUAGE MODELS

Native VLMs embrace early-fusion integration rather than grafting VEs onto LLMs. Early Fuyu (Bavishi et al., 2023), EVE (Diao et al., 2024), and SOLO (Chen et al., 2024d), embed image patches via linear projections, whereas Chameleon (Team, 2024), MoMA (Lin et al., 2024), and MoT (Liang et al., 2024) transform images into symbolic sequences via discrete tokenizers. Later studies (Luo et al., 2024; Diao et al., 2025; Li et al., 2025b; Luo et al., 2025; Li et al., 2025a) leverage Mixture-of-Experts (MoE) or Divide-and-Conquer (DaC) strategies to suppress vision-language interference, while others (Diao et al., 2024; Li et al., 2025b; Wang et al., 2025a; Li et al., 2025a) upgrade visual encoder supervision to accelerate the acquisition of visual concepts. Empirical evidence (Beyer et al., 2024; Luo et al., 2024; Lei et al., 2025) reveals that, with sufficient data and progressive training, native VLMs rapidly approach modular counterparts, corroborating recent scaling-law insights (Shukor et al., 2025b;a). Besides, recent methods (Tao et al., 2025; Yan et al., 2025; Xiao et al., 2025) indicate that multi-modality encoding modules with the LLM or VE style slightly resolve vision-language misalignment, yet fail to fully integrate the distinct properties of each modality.

Notably, NEO redefines native VLMs as a unibody system built from first-principle primitives. Every network component, from native rotary position embeddings to multi-modality interaction patterns, ensures full compatibility with the intrinsic modeling patterns of VEs and LLMs. Meanwhile, NEO differs from existing modular VLMs via modality-agnostic pre-Buffer and end-to-end training, dramatically enhancing pixel-word alignment and pushing the frontier of native VLM research.

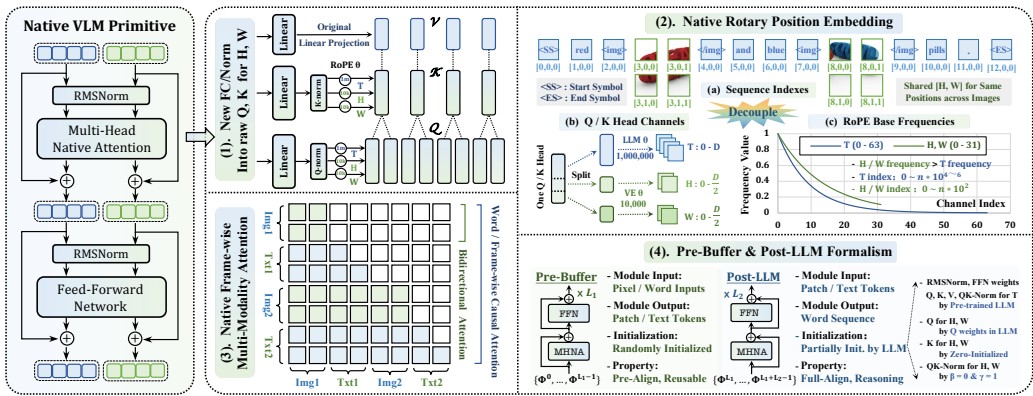

Figure 3: Overview of our native primitive, which integrates native attention with bi-directional dependencies within images and word / frame-wise causal interactions, together with native rotary position embeddings parameterized by modality-specific frequency, channel, and index allocation. It is inherently unified and intrinsically multimodal, substantially enhancing pixel–word correspondence.

## 3 METHODOLOGY

### 3.1 MODEL ARCHITECTURE

Figure 2 illustrates the proposed NEO framework, which comprises lightweight patch and word embedding layers, a pre-Buffer, and a post-LLM, built upon stacked native VLM primitives.

**Patch and Word Embeddings.** Given an image $\mathbf{I}$, we convert it into token sequences via a lightweight Patch Embedding Layer (PEL) with two Convolutional layers (Conv1–2) (Krizhevsky et al., 2012) and a Gaussian Error Linear Unit (GELU) (Hendrycks & Gimpel, 2016). For input text $\mathbf{T}$, we encode it into word tokens using the original LLM Tokenizer as Word Embedding Layer (WEL):

$$\boldsymbol{x_v} = \text{Conv2}(\text{GELU}(\text{Conv1}(\mathbf{I})) + \mathbf{PE}), \quad \boldsymbol{x_t} = \text{Tokenizer}(\mathbf{T}), \tag{1}$$

where $\boldsymbol{x_v} \in \mathbb{R}^{(h \times w) \times d}$ / $\boldsymbol{x_t} \in \mathbb{R}^{n \times d}$ denote visual / textual tokens, and $\mathbf{PE}$ is 2D Sinusoidal Positional Encoding (Dosovitskiy et al., 2021). The stride of Conv1 / Conv2 is 16 / 2, *i.e.*, each visual token corresponds to a $32 \times 32$ image patch. Notably, Conv2 performs token folding like pixel unshuffle (Chen et al., 2024e), with the special  and </img> tokens inserted at the boundaries of visual tokens, while mapping position and patch embeddings into a unified space. Afterward, visual and textual tokens are merged and propagated through the unified backbone.

**Native VLM Primitive.** It adopts RMSNorm (Zhang & Sennrich, 2019) and SwiGLU (Dauphin et al., 2017) consistent with LLM layers. Unlike prior methods that collapse visual tokens into 1D representations (Zhu et al., 2025; Wang et al., 2025b) or merely reallocate pre-trained LLM head dimensions across temporal (T), height (H), and width (W) (Wang et al., 2024a; Bai et al., 2025), we enlarge Query (Q) and Key (K) head dimensions and decouple H, W, and T relations in Figure 3(1), adding ~10% more parameters over the raw Transformer block. The T dimension is retained, and new H and W dimensions are added with their respective QK normalization (Yang et al., 2025).

This philosophy aligns with our **Native Rotary Position Embedding (Native-RoPE)** in Figure 3(2). *(a) Index Allocation*: For text, T index is retained while H / W indexes are zeroed. For images, each visual token has a constant T index, with unique H / W indexes encoding spatial location. Videos, treated as sequences of frames, increment T index per frame, while H / W indexes follow the same spatial scheme as images. In multimodal inputs, each modality's T index starts from the maximum ID of the preceding modality, ensuring continuous and unambiguous positional encoding across modalities. This serves two purposes: (-) For image pairs, H / W indexes start independently from (0,0), and tokens at corresponding positions share identical dependency, strongly reinforcing correlations and interactions across matching regions (Liao et al., 2025; Wu et al., 2025); (-) For image-text pairs, H / W indexes are decoupled from T index and bounded within (0,0) and (H,W), preventing large T index growth from disproportionately affecting H / W indexes (Wang et al., 2024a; Bai et al., 2025) and thereby keeping spatial dependencies between long-range text and images.

Another key aspect is *(b) Channel and (c) Frequency Allocation*. Unlike recent 3D-RoPE methods (Bai et al., 2025; Wei et al., 2025; Yuan et al., 2025; Liao et al., 2025), we fully decompose the channel and frequency allocation of H, W, and T, equipped with additional Q/K head dimensions for H and W. This resolves two issues: (-) Zeroing H / W indexes for pure text would disrupt the modeling patterns and linguistic capacity of the LLM if restricted to its original channels. Repairing this disruption requires substantial resources; (-) Even with interleaved or segmented reallocation, H and W are theoretically equivalent but are assigned different frequencies. Meanwhile, the RoPE frequency in LLMs is far lower than that of visual encoders in Figure 3(2c). This mismatch limits the modeling of relative distances and local semantics. The problem is exacerbated by the disparity in scales, with temporal ranges spanning up to one million and spatial ranges only a few hundred.

Specifically, Native-RoPE assigns distinct base frequencies to T, H, and W within their own dimensions, *i.e.*, original LLM head dimension for T and new head dimension for H / W as follows:

$$\Theta_T = \left\{ \beta_T^{-\frac{2k}{d}} \mid k \in [0, \frac{d}{2}) \right\}, \ \Theta_H = \left\{ \beta_H^{-\frac{4i}{d}} \mid i \in [0, \frac{d}{4}) \right\}, \ \Theta_W = \left\{ \beta_W^{-\frac{4j}{d}} \mid j \in [0, \frac{d}{4}) \right\} \quad (2)$$

where $\beta$ and $\Theta$ indicate the base and rotation frequency across H, W, and T. Notably, temporal T dimension captures both local and long-range relations, whereas spatial H / W dimensions emphasize local dependencies. This also opens avenues for broader applications, *e.g.*, video understanding (Wei et al., 2025), multimodal generation (Deng et al., 2025b), and editing (Deng et al., 2025a).

Inspired by prior works (Lei et al., 2025; Deng et al., 2025b; Li et al., 2025a; Beyer et al., 2024), we also treat one single image as a unified meta-unit for autoregressive modeling, denoted as **Native Multi-Modal Attention** with mixed masking in Figure 3(3). Text tokens adhere to standard causal attention, attending only to preceding tokens to maintain autoregressive generation. In contrast, image tokens employ full bidirectional attention, enabling exhaustive interactions among all visual tokens, akin to a visual encoder. This design captures rich spatial and contextual dependencies within images and facilitates vision-language correspondences, thereby supporting complex multimodal reasoning. We use FlexAttention (Dong et al., 2024) to minimize memory overhead and increase throughput, as variable-length block-wise attention is fully optimized through CUDA kernel modifications.

**Pre-Buffer and Post-LLM.** In Figure 3(4), we develop a modality-shared pre-Buffer to translate pixel–word inputs into a unified representation with minimal disturbance to the post-LLM, which inherits the linguistic proficiency and reasoning capabilities of pre-trained LLM. The layer depths $L_1$ and $L_2$ primarily refer to parameter counts and scaling properties (Tian et al., 2025) of existing VEs and LLMs to balance accuracy and efficiency. Here, we formulate one primitive $\Phi^l$ as follows:

$$\boldsymbol{x}_m^{l'} = \boldsymbol{x}_m^l + \text{MHNA}(\text{RMSNorm}(\boldsymbol{x}_m^l)), \quad \boldsymbol{x}_m^{l+1} = \boldsymbol{x}_m^{l'} + \text{FFN}(\text{RMSNorm}(\boldsymbol{x}_m^{l'})), \quad (3)$$

where $\boldsymbol{m} \in \{\boldsymbol{v}, \boldsymbol{t}\}$ indicates input modality. Besides, $\{\Phi^0, ..., \Phi^{L_1-1}\}$ and $\{\Phi^{L_1}, ..., \Phi^{L_1+L_2-1}\}$ denotes pre-Buffer and post-LLM, respectively. Notably, we randomly initialize the entire pre-Buffer, while the post-LLM inherits RMSNorm, Feed-Forward Network (FFN), and Q/K/QK-Norm parameters along the temporal dimension from a pretrained LLM. The temporal Q is reused to initialize Q for the H and W dimensions, their K weights are zero-initialized, and the corresponding QK-Norm is initialized with $\beta = 0$ and $\gamma = 1$. We further match the attention scaling to that of the pretrained LLM, thereby preserving its pre-training paradigm from the outset and enabling a progressive emergence of multimodal spatial reasoning within the post-LLM. Crucially, this separation exists only during pre-training. After that, these components are merged into a monolithic backbone that autonomously allocates capacity for encoding, alignment, and reasoning.

### 3.2 TRAINING PROCEDURE

Figure 4 illustrates the whole training pipeline, where the entire model is optimized end-to-end.

**Pre-Training Stage.** In this phase, NEO acquires fundamental visual concepts and contextual dependencies from scratch, guided by pre-trained patterns from LLMs. Training leverages 345M web-scale and synthetic image-caption pairs, including 100M English and 20M Chinese pairs from LAION-400M (Schuhmann et al., 2021), 150M English pairs from COYO-700M (Byeon et al., 2022), 20M long-caption examples from BLIP3o (Chen et al., 2025), and 5M short-caption pairs from OpenImages (Kuznetsova et al., 2018), recaptioned with a pre-trained InternVL2-8B model. The dataset is further enriched with 30M samples from LAION-COCO (Schuhmann et al., 2022)

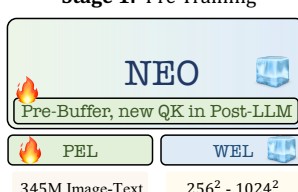
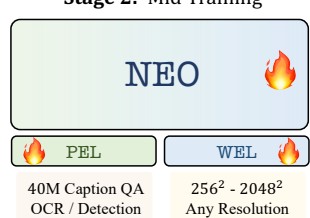
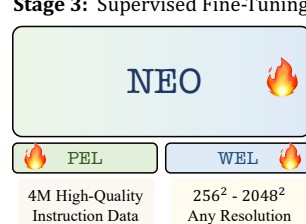

Figure 4: Overview of the entire training recipe. During pre-training, NEO learns visual perception from massive web-scale and synthetic image-caption pairs with frozen LLM weights to preserve linguistic knowledge. In mid-training and supervised fine-tuning, the full model is progressively optimized end-to-end using caption, conversation, OCR, detection, and high-quality instruction data.

and 20M examples from Wukong (Gu et al., 2022) with rich Optical Character Recognition (OCR) annotations. A 3:7 ratio of language to multi-modal data is incorporated to reconstruct text projections in the pre-Buffer. Only the patch embedding layer, the pre-Buffer, and extra QK linear weights and normalization in post-LLM, along with H and W, are optimized with a simple next-token prediction objective. Notably, the new QK heads not only counteract the LLM's strong language bias that limits visual specialization but also safeguard its capabilities against the effects of low-quality data.

**Mid-Training Stage.** The objective at this stage is to strengthen the alignment between visual and linguistic capabilities while progressively enhancing recognition of high-resolution images, complex scenes, object scales, spatial grounding, and compact OCR content. The training data is drawn from the pre-training corpus of InternVL-1.5 (Chen et al., 2024f), comprising 40M samples across image captioning, conversation, detection, and OCR data, which account for approximately 66%, 11%, 8%, and 15% of the total, respectively. A 3:7 ratio of language to multi-modal data is again applied. The entire architecture is updated with the same loss functions to consolidate vision-language alignment, thereby equipping NEO with the foundational abilities required for various visual scenarios.

**Supervised Fine-Tuning Stage.** During the SFT stage, NEO's ability to follow complex linguistic instructions and varied dialogue patterns is further enhanced, a critical step towards real-world deployment. The full network is optimized across diverse high-quality, multi-source instruction datasets. Following Mono-InternVL (Luo et al., 2024), we employ about 4M bilingual instructions for supervised learning, covering tasks such as visual question answering, multimodal dialogue, mathematics, and knowledge reasoning. Details of the instruction data are provided in the Appendix.

## 4 EXPERIMENTS

### 4.1 TRAINING SETTINGS

Our NEO models are built on Qwen3-1.7B and Qwen3-8B (Yang et al., 2025) as the LLMs. The pre-Buffer employs $L_1 = 12$ primitive layers for NEO-2.2B and $L_1 = 6$ for NEO-9B. We extend only the QK head dimension in raw transformer layers, introducing roughly 10% extra parameters over the original design. The base RoPE frequencies $\beta_T$, $\beta_H$, and $\beta_W$ are set to $1 \times 10^6$, $1 \times 10^4$, and $1 \times 10^4$, respectively. NEO is trained on sixteen 8-GPU (80G) nodes using the AdamW optimizer (Loshchilov & Hutter, 2019). The maximum learning rates for pre-training, mid-training, and SFT are $8 \times 10^{-4}$, $4 \times 10^{-5}$, and $5 \times 10^{-5}$, with a warm-up ratio of 0.01 and a cosine decay scheduler across all stages.

### 4.2 MAIN RESULTS

We conduct standard evaluations with VLMEvalKit (Duan et al., 2024) on diverse benchmarks, covering chart, diagram, and document understanding tasks, *e.g.*, AI2D (Kembhavi et al., 2016), DocVQA (Clark & Gardner, 2018), ChartQA (Masry et al., 2022), InfoVQA (Mathew et al., 2022), TextVQA (Singh et al., 2019), and OCRBench (Liu et al., 2023e); visual perception and challenging reasoning tasks, *e.g.*, MMMU (Yue et al., 2024), MMBench-EN (MMB) (Liu et al., 2024b), MMVet (Yu et al., 2024), MMStar (Chen et al., 2024c), SEEDBench-IMG (SEED-I) (Li et al., 2023a); hallucination tasks, *e.g.*, POPE (Li et al., 2023b) and HallusionBench (HallB) (Guan et al., 2024).

Table 1: **Comparison with modular and native VLMs on general vision-language benchmarks.** "# Data" denotes the dataset scale during pre-training, mid-training, and supervised fine-tuning. [†] indicates models that employ reinforcement learning (RL). **Bold** highlights the highest performance.

| Model | LLM | # Data | MMMU | MMB | MMVet | MMStar | SEED-I | POPE | HallB |
|---|---|---|---|---|---|---|---|---|---|
| ▼ *Modular Vision-Language Models (2B)* | | | | | | | | | |
| Qwen2-VL | Qwen2-1.5B | − / − / − | 41.1 | 74.9 | 49.5 | 48.0 | − | − | 41.7 |
| InternVL2.5 | InternLM2.5-1.8B | >6B / 100M / 16M | 43.6 | 74.7 | 60.8 | 53.7 | − | **90.6** | 42.6 |
| InternVL3[†] | Qwen2.5-1.5B | >6B / 100M / 22M | **48.6** | **81.1** | 62.2 | 60.7 | − | 89.6 | 42.5 |
| Qwen2.5-VL[†] | Qwen2.5-3B | − / − / − | 51.2 | 79.1 | 61.8 | 55.9 | − | − | 46.3 |
| Encoder-Based | Qwen3-1.7B | >6B / 40M / 4M | 47.1 | 75.8 | 37.4 | 52.7 | **73.6** | 87.0 | **44.4** |
| ▼ *Native Vision-Language Models (2B)* | | | | | | | | | |
| Mono-InternVL | InternLM2-1.8B | 1.2B / 143M / 7M | 33.7 | 65.5 | 40.1 | − | 67.4 | − | 34.8 |
| Mono-InternVL-1.5 | InternLM2-1.8B | 400M / 150M / 7M | 39.1 | 64.0 | **54.0** | − | 66.9 | − | 32.5 |
| HoVLE | InternLM2-1.8B | 550M / 50M / 7M | 32.2 | 73.3 | 43.8 | − | 70.9 | 87.4 | 38.4 |
| OneCAT | Qwen2.5-1.5B | 436M / 70M / 13M | 39.0 | 72.4 | 42.4 | − | 70.9 | − | − |
| NEO | Qwen3-1.7B | 345M / 40M / 4M | **48.6** | **76.0** | 49.6 | **54.2** | **74.2** | **87.5** | **43.1** |
| ▼ *Modular Vision-Language Models (8B)* | | | | | | | | | |
| Qwen2-VL | Qwen2-7B | − / − / − | 54.1 | 83 | 62.0 | 60.7 | − | 88.1 | 50.6 |
| InternVL2.5 | InternLM2.5-7B | >6B / 50M / 4M | 56.0 | **84.6** | 62.8 | 64.4 | − | 90.6 | 50.1 |
| Qwen2.5-VL[†] | Qwen2.5-7B | − / − / − | 55.0 | 83.5 | 67.1 | 63.9 | − | 86.4 | **52.9** |
| InternVL3[†] | Qwen2.5-7B | >6B / 100M / 22M | **62.7** | 83.4 | **81.3** | **68.2** | − | **91.1** | 49.9 |
| Encoder-Based | Qwen3-8B | >6B / 40M / 4M | 54.1 | 84 | 60.0 | 63.5 | **76.2** | 87.8 | 51.4 |
| ▼ *Native Vision-Language Models (8B)* | | | | | | | | | |
| Fuyu | Persimmon-8B | − / − / − | 27.9 | 10.7 | 21.4 | − | 59.3 | 84.0 | − |
| Chameleon | from scratch | 1.4B / 0M / 1.8M | 25.4 | 31.1 | 8.3 | − | 30.6 | 19.4 | 17.1 |
| EVE | Vicuna-7B | 33M / 0M / 1.8M | 32.6 | 52.3 | 25.7 | − | 64.6 | 85.0 | 26.4 |
| SOLO | Mistral-7B | 44M / 0M / 2M | − | 67.7 | 30.4 | − | 64.4 | 78.6 | − |
| Emu3 | from scratch | − / − / − | 31.6 | 58.5 | 37.2 | − | 68.2 | 85.2 | − |
| EVEv2 | Qwen2.5-7B | 77M / 15M / 7M | 39.3 | 66.3 | 45.0 | − | 71.4 | 87.6 | − |
| BREEN | Qwen2.5-7B | 13M / 0M / 4M | 42.7 | 71.4 | 38.9 | 51.2 | − | − | 37.0 |
| VoRA | Qwen2.5-7B | 30M / 0M / 0.6M | 32.0 | 61.3 | 33.7 | − | 68.9 | 85.5 | − |
| SAIL | Mistral-7B | 512M / 86M / 6M | − | 70.1 | 46.3 | 53.1 | 72.9 | 85.8 | **54.2** |
| NEO | Qwen3-8B | 345M / 40M / 4M | **54.6** | **82.1** | 53.6 | **62.4** | **76.3** | **88.4** | 46.4 |

Following InternVL3 (Zhu et al., 2025), we construct the *Encoder-Based* by combining Qwen3 (Yang et al., 2025) and InternViT-300M (Zhu et al., 2025). In the mid-training stage, we first train the projector on 10M samples, and further unfreeze the vision encoder utilizing another 30M samples.

**Comparison with Modular VLMs.** In Table 1 and Table 2, NEO achieves highly competitive performance against Encoder-Based counterparts at the 2B and 8B scales. Impressively, NEO largely narrows the performance gap with top-tier modular VLMs, *e.g.*, Qwen2-VL (Wang et al., 2024a), InternVL2.5 (Chen et al., 2024e), Qwen2.5-VL (Bai et al., 2025), and InternVL3 (Zhu et al., 2025) across multiple benchmarks, despite using relatively limited training data and without reinforcement learning. These results highlight the effectiveness of an end-to-end training strategy and a unified model design with Native-RoPE and multi-modality interaction patterns. Moreover, the performance gap between Encoder-Based variants and state-of-the-art methods on MMMU, MMVet, TextVQA, and *etc*, indicates that NEO still suffers from the limitation in training data scale and quality.

**Comparison with Native VLMs.** From Table 1 and Table 2, NEO delivers substantial gains on visual-centric benchmarks over the best competitors, *e.g.*, Mono-InterVL (Luo et al., 2024; 2025), HoVLE (Tao et al., 2025), OneCAT (Li et al., 2025a), EVE (Diao et al., 2024; 2025), Emu3 (Wang et al., 2024b), BREEN (Li et al., 2025b), VoRA (Wang et al., 2025a), and SAIL (Lei et al., 2025). By seamlessly integrating post-LLM components with the pre-Buffer for large-scale visual learning, NEO aligns visual inputs with textual features from scratch and supports complex visual reasoning, even without visual encoder supervision (Diao et al., 2024; Tao et al., 2025; Li et al., 2025a; Wang et al., 2025a; Li et al., 2025b), highlighting the strengths of its native primitive designs and training strategies. These design choices allow NEO to surpass many native VLMs using fewer training resources, demonstrating the advantages of our primitives with efficient data-scaling capability.

Table 2: **Comparison with modular and native VLMs on visual question answering benchmarks.** Any Res., Tile-wise, Any Rat., and Fix Res. refer to any resolution, image tile splitting, any aspect ratio, and fixed resolution. MoE and DaC are Mixture-of-Experts and Divide-and-Conquer models.

| Model | Input | RoPE | Backbone | AI2D | DocVQA | ChartQA | InfoVQA | TextVQA | OCRBench |
|---|---|---|---|---|---|---|---|---|---|
| ▼ *Modular Vision-Language Models (2B)* | | | | | | | | | |
| Qwen2-VL | Any Res. | M-RoPE | Dense | 74.7 | **90.1** | 73.5 | 65.5 | **79.7** | 80.9 |
| InternVL2.5 | Tile-wise | 1D-RoPE | Dense | 74.9 | 88.7 | 79.2 | 60.9 | 74.3 | 80.4 |
| InternVL3[†] | Tile-wise | 1D-RoPE | Dense | **78.7** | 88.3 | **80.2** | **66.1** | 77.0 | **83.5** |
| Qwen2.5-VL[†] | Any Res. | M-RoPE | Dense | 81.6 | 93.9 | 84.0 | 77.1 | 79.3 | 79.7 |
| Encoder-Based | Tile-wise | 1D-RoPE | Dense | 77.4 | 89.9 | 78.4 | 65.9 | 73.3 | **83.5** |
| ▼ *Native Vision-Language Models (2B)* | | | | | | | | | |
| Mono-InternVL | Tile-wise. | 1D-RoPE | MoE | 68.6 | 80.0 | 73.7 | 43.0 | 72.6 | 76.7 |
| Mono-InternVL-1.5 | Tile-wise. | 1D-RoPE | DaC | 67.4 | 81.7 | 72.2 | 47.9 | 73.7 | **80.1** |
| HoVLE | Tile-wise. | 1D-RoPE | Dense | 73.0 | 86.1 | 78.6 | 55.7 | 70.9 | 74.0 |
| OneCAT | Any Res. | M-RoPE | Dense | 72.4 | 87.1 | 76.2 | 56.3 | 67.0 | – |
| NEO | Any Res. | Native-RoPE | Dense | **80.1** | **89.9** | **81.2** | **63.2** | **74.0** | 77.1 |
| ▼ *Modular Vision-Language Models (8B)* | | | | | | | | | |
| Qwen2-VL | Any Res. | M-RoPE | Dense | 83.0 | 94.5 | 83 | 76.5 | 84.3 | 86.6 |
| InternVL2.5 | Tile-wise | 1D-RoPE | Dense | 84.5 | 93.0 | 84.8 | 77.6 | 79.1 | 82.2 |
| Qwen2.5-VL[†] | Any Res. | M-RoPE | Dense | 83.9 | **95.7** | **87.3** | **82.6** | **84.9** | 86.4 |
| InternVL3[†] | Tile-wise | 1D-RoPE | Dense | **85.2** | 92.7 | 86.6 | 76.8 | 80.2 | **88** |
| Encoder-Based | Tile-wise | 1D-RoPE | Dense | 82.9 | 92.1 | 83.5 | 75 | 77.1 | 85.3 |
| ▼ *Native Vision-Language Models (8B)* | | | | | | | | | |
| Fuyu | Any Res. | 1D-RoPE | Dense | 64.5 | – | – | – | – | 36.6 |
| Chameleon | Fix Res. | 1D-RoPE | Dense | 46.0 | 1.5 | 2.9 | 5.0 | 4.8 | 0.7 |
| EVE | Any Rat. | 1D-RoPE | Dense | 61.0 | 53.0 | 59.1 | 25.0 | 56.8 | 39.8 |
| SOLO | Any Res. | 1D-RoPE | Dense | 61.4 | – | – | – | – | 12.6 |
| Emu3 | Fix Res. | 1D-RoPE | Dense | 70 | 76.3 | 68.6 | 43.8 | 64.7 | 68.7 |
| EVEv2 | Any Rat. | 1D-RoPE | DaC | 74.8 | – | 73.9 | – | 71.1 | 70.2 |
| BREEN | Any Res. | 1D-RoPE | MoE | 76.4 | – | – | – | 65.7 | – |
| VoRA | Any Res. | 1D-RoPE | Dense | 61.1 | – | – | – | 58.7 | – |
| SAIL | Any Res. | M-RoPE | Dense | 76.7 | – | – | – | **77.1** | **78.3** |
| NEO | Any Res. | Native-RoPE | Dense | **83.1** | **88.6** | **82.1** | **60.9** | 75.0 | 77.7 |

Despite strong results, NEO lags on knowledge-/OCR-heavy tasks, *e.g.*, MMMU, InfoVQA, and TextVQA. *Interestingly, NEO-9B does not surpass NEO-2B on DocVQA and InfoVQA*, indicating limitations in our current training corpus. Even so, NEO performs well under constraints, highlighting the native VLM as a scalable paradigm. Larger datasets and resources can unlock its full potential.

## 4.3 ABLATION STUDIES

Unless otherwise specified, we report the average evaluation results, denoted as **Avg.**, across ten vision-language benchmark datasets in Table 3. The pre-Buffer and new head dimensions in the post-LLM are trained on 20M pre-training samples, followed by full-backbone fine-tuning on 2M SFT instruction data. These constitute the standard training settings for our ablation studies.

**Hyperparameters of the Pre-Buffer Layer.** Figure 5 illustrates the relationship between the number of pre-Buffer layers and the model's average accuracy, using Qwen3-1.7B as the post-LLM. Performance improves consistently as the layer count increases, but gains begin to saturate beyond eight layers. Inspired by the parameter counts of popular VEs (Chen et al., 2024f; Radford et al., 2021; Zhai et al., 2023) and the scaling properties (Tian et al., 2025) between existing VEs and LLMs, we select 12 and 6 layers for NEO-2.2B and NEO-9B to balance the trade-off between performance and efficiency.

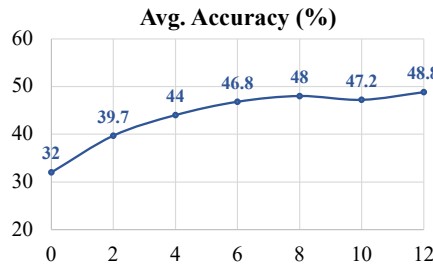

Figure 5: Configurations of pre-Buffer.

Table 3: Configurations of attention and RoPE. MMS, CQA, IVQA, and OCRB denote MMStar, ChartQA, InfoVQA, and OCRBench. ⋆ indicates that the base RoPE frequencies for height and width are set to 1M. To ensure fairness, we add new head dimensions of equal size across all models.

| Model | Attention | RoPE | MMMU | MMB | MMS | SEED-I | AI2D | CQA | IVQA | TVQA | OCRB | POPE | Avg. |
|---|---|---|---|---|---|---|---|---|---|---|---|---|---|
| A | Causal | 1D-RoPE | 40.2 | 48.6 | 36.1 | 55.3 | 63.6 | 16.1 | 22.5 | 16.2 | 13.9 | 78.6 | 39.1 |
| B | Mixed | 1D-RoPE | **40.8** | 48.8 | 36.4 | 57.3 | 63.7 | 16.0 | 21.9 | 17.4 | 16.0 | 79.2 | 39.8 |
| C | Mixed | IL-RoPE | 40.0 | 47.3 | 36.3 | 57.6 | 62.0 | 18.8 | 23.4 | 17.9 | 13.2 | 78.8 | 39.5 |
| D | Mixed | M-RoPE | 40.3 | 49.6 | 37.2 | 57.8 | 64.2 | 23.7 | 25.2 | 20.4 | 18.8 | 79.3 | 41.7 |
| E | Mixed | MM-RoPE | 40.5 | 50.8 | 37.6 | 58.2 | **65.8** | 25.7 | **26.3** | 22.1 | 18.2 | 78.8 | 42.4 |
| F | Mixed | Video-RoPE | 40.6 | **51.3** | **37.8** | **58.8** | 64.3 | **27.4** | 26.1 | **23.7** | **21.3** | **81.0** | **43.2** |
| G | Causal | Native-RoPE | 40.2 | 49.2 | 36.3 | 57.1 | 63.7 | 19.2 | 23.5 | 19.5 | 16.7 | 77.8 | 40.3 |
| H | Mixed | Native-RoPE | **40.7** | **51.9** | **38.2** | **58.9** | **65.8** | **30.6** | **26.9** | **24.1** | **23.2** | 80.0 | **44.0** |
| I | Mixed | Native-RoPE⋆ | 40.4 | 50.4 | 36.9 | 57.0 | 64.1 | 25.6 | 25.2 | 21.7 | 20.1 | 78.7 | 42.0 |

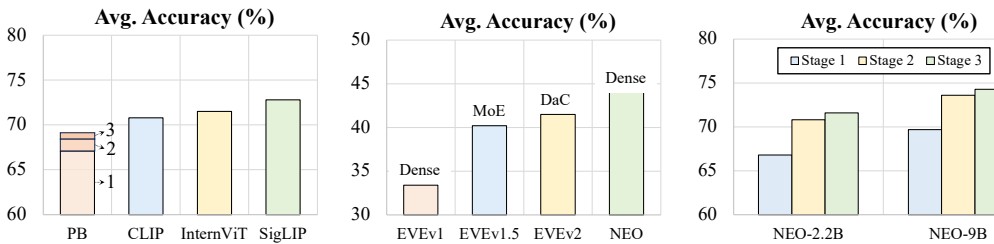

Figure 6: Pre-Buffer vs. VEs.  Figure 7: Neo vs. EVE series.  Figure 8: Three training processes.

**Configurations of Native Primitives.** Table 3 compares various attention and RoPE designs. The pre-Buffer depth is 4, and the post-LLM is initialized with Qwen3-1.7B. All models share the same new QK head dimensions and normalization. *(1) Attention mode.* Comparing models A/B and G/H reveals consistent gains of mixed attention over causal one, reflecting its stronger capacity to model comprehensive dependencies and cross-modal alignment. *(2) RoPE mode.* Native-RoPE outperforms 1D-RoPE (Zhu et al., 2025), IL-RoPE (Liao et al., 2025), M-RoPE (Bai et al., 2025), MM-RoPE (Yuan et al., 2025), and Video-RoPE (Wei et al., 2025), with at least a 0.8% gain. This validates the importance of disentangling height, width, and temporal components in RoPE to enhance spatial–temporal representations and fine-grained interactions. By contrast, setting the base RoPE frequency to 1M for height and width severely impairs the ability to perceive local semantics.

**Comparison between Pre-Buffer and Vision Encoders.** In Figure 6, PB 1–3 denotes the Pre-Buffer obtained from stage 1–3. For fairness, all post-LLMs are initialized by Qwen3-1.7B (Yang et al., 2025) combined by pre-trained pre-Buffer, CLIP-vit-large-patch14 (Radford et al., 2021), InternViT-300M (Chen et al., 2024e), and SigLIP-so400m-patch14-384 (Zhai et al., 2023). During pre-training, we train the projector for encoder-based methods and the newly added QK parameters for all models. During SFT, we unfreeze the entire backbone. After two-stage re-training, PB3 delivers strong performance, trailing CLIP / InternViT / SigLIP by only 1.7 / 2.4 / 3.7% on average across diverse benchmarks. Notably, despite using only 22M training samples, PB3 is just 2.5% below the full NEO model, substantially reducing the training cost of developing native VLMs for future research.

**Comparison between NEO and Native VLM variants.** Existing native VLMs in Table 1 and 2 differ dramatically in training pipelines, data volume/quality, and base LLM choice. To isolate model architectural effects, we compare our NEO with representative EVEv1.0 (Dense), EVEv1.5 (Mixture-of-Experts), and EVEv2.0 (Divide-and-Conquer), all built on Qwen3-1.7B. From Figure 7, EVEv1.0, EVEv1.5, and EVEv2.0 achieves 33.4%, 40.2%, and 41.5%, respectively, compared with 44.0% for our NEO. This confirms that the gains stem from our pre-Buffer, native RoPE design, and interaction patterns, rather than from a newer backbone or larger and higher-quality data alone.

**Performance Gains across Stages.** Figure 8 presents the result evolution across training stages. In Stages 1 and 2, the model is fine-tuned on 2M SFT examples. Performance improves consistently as training data scales increase across 2.2B and 9B model sizes. Following progressive training, NEO shows strong multimodal capabilities, enabling robust performance across diverse real-world tasks.

## 5    CONCLUSION

We introduce NEO, a native VLM that seamlessly integrates vision and language into a single unified framework, eliminating the need for separate visual encoders or ad-hoc alignment modules. By leveraging hybrid attention and modality-aware rotary position embeddings, NEO captures rich, fine-grained interactions between pixels and words from the outset. Its pre-Buffer and post-LLM training paradigm ensures efficient convergence and robust alignment while maintaining end-to-end learning. Experiments show that this unified design not only advances multimodal understanding and reasoning but also lays the foundation for reusable, scalable components. Our native primitives highlight a promising path toward intrinsically multimodal, unified, and adaptable architectures.

## ETHICS STATEMENT

All resources are drawn from open-access datasets with explicitly defined usage policies. Our work seeks to advance multimodal learning capabilities without introducing ethical or safety concerns beyond those already associated with existing models. Nevertheless, risks such as dataset biases and potential misuse cannot be entirely ruled out. We emphasize the importance of careful data curation, responsible deployment, and transparent reporting as essential practices to mitigate these challenges.

## REPRODUCIBILITY STATEMENT

We place strong emphasis on reproducibility, providing detailed descriptions to facilitate replication and validation. Information about dataset selection, training strategies, and evaluation settings is provided in Sec. 3.2 and Sec. 4.1. We commit to releasing the code, model weights, and detailed documentation to allow the community to reproduce our findings in future research.

## ACKNOWLEDGEMENT

This research is supported by cash and in-kind funding from NTU S-Lab and industry partner(s). This study is also supported by the Ministry of Education, Singapore, under its MOE AcRF Tier 2 (MOE-T2EP20221-0012, MOE-T2EP20223-0002).

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

## A APPENDIX

### USAGE OF LARGE LANGUAGE MODELS

During manuscript preparation, large language models were used solely as writing assistants. They helped to check grammar, refine sentence structure, and provide style alternatives. All content related to methodology, experiments, and conclusions was developed entirely by the authors. LLM outputs were reviewed critically, and only human-verified edits were incorporated into the final text.

### A.1 SUPERVISED FINE-TUNING DATASETS

Table 4: Dataset summary in supervised fine-tuning stage.

| Task | Dataset |
|---|---|
| Captioning | TextCaps (en) (Sidorov et al., 2020), ShareGPT4V (en&zh) (Chen et al., 2024b) |
| General QA | VQAv2 (en) (Goyal et al., 2017), GQA (en) (Hudson & Manning, 2019), OKVQA (en) (Marino et al., 2019), VSR (en) (Liu et al., 2023a), VisualDialog (en) (Das et al., 2017) |
| Science | AI2D (en) (Kembhavi et al., 2016), ScienceQA (en) (Lu et al., 2022a), TQA (en) (Kembhavi et al., 2017) |
| Chart | ChartQA (en) (Masry et al., 2022), MMC-Inst (en) (Liu et al., 2023c), DVQA (en) (Kafle et al., 2018), PlotQA (en) (Methani et al., 2020), LRV-Instruction (en) (Liu et al., 2023b) |
| Mathematics | GeoQA+ (en) (Cao & Xiao, 2022), TabMWP (en) (Lu et al., 2022b), MathQA (en) (Yu et al., 2023), CLEVR-Math/Super (en) (Lindström & Abraham, 2022; Li et al., 2023c), Geometry3K (en) (Lu et al., 2021) |
| Knowledge | KVQA (en) (Shah et al., 2019), A-OKVQA (en) (Schwenk et al., 2022), ViQuAE (en) (Lerner et al., 2022), Wikipedia (en&zh) (He et al., 2023) |
| OCR | OCRVQA (en) (Mishra et al., 2019), InfoVQA (en) (Mathew et al., 2022), TextVQA (en) (Singh et al., 2019), ArT (en&zh) (Chng et al., 2019), COCO-Text (en) (Veit et al., 2016), CTW (zh) (Yuan et al., 2019), LSVT (zh) (Sun et al., 2019), RCTW-17 (zh) (Shi et al., 2017), ReCTs (zh) (Liu et al., 2019), SynthDoG (en&zh) (Kim et al., 2022), ST-VQA (en) (Biten et al., 2019) |
| Document | DocVQA (en) (Clark & Gardner, 2018), Common Crawl PDF (en&zh) |
| Grounding | RefCOCO/+/g (en) (Yu et al., 2016; Mao et al., 2016), Visual Genome (en) (Krishna et al., 2017) |
| Conversation | LLaVA-150K (en&zh) (Liu et al., 2023d), LVIS-Instruct4V (en) (Wang et al., 2023), ALLaVA (en&zh) (Chen et al., 2024a), Laion-GPT4V (en) (LAION, 2023), TextOCR-GPT4V (en) (Jimmycarter, 2023), SVIT (en&zh) (Zhao et al., 2023) |
| Text-only | OpenHermes2.5 (en) (Teknium, 2023), Alpaca-GPT4 (en) (Taori et al., 2023), COIG-CQIA (zh) (Bai et al., 2024), ShareGPT (en&zh) (Zheng et al., 2023) |

### A.2 IMPLEMENTATION DETAILS

Table 5: Implementation details in the pre-training, mid-training and supervise fine-tuning.

| Configuration | Pre-Training | Mid-Training | Supervised Fine-Tuning |
|---|---|---|---|
| Resolution | $256^2 - 1,024^2$ | $256^2 - 2,048^2$ | $256^2 - 2,048^2$ |
| Optimizer | | AdamW | |
| Optimizer hyperparameters | | $\beta_1 = 0.9, \quad \beta_2 = 0.999, \quad eps = 1e^{-8}$ | |
| Learning rate schedule | cosine with min lr | cosine with min lr | cosine decay |
| Peak learning rate | $8e^{-4}$ | $4e^{-5}$ | $5e^{-5}$ |
| Min learning rate ratio | 0.05 | 0.1 | – |
| Weight decay | | 0.01 | |
| Training steps | 190k | 50k | 6k |
| Warm-up steps | 2k | 200 | 200 |
| Max sample length | $8,192$ | $8,192$ | $8,192$ |
| Global batch size | $2,560$ | $1,200$ | 650 |
| Text-only ratio | 0.3 | 0.3 | – |
| Numerical precision | | `bfloat16` | |

### A.3 LIMITATION AND DISCUSSION

In this study, we innovate network architectures and training strategies for efficiently building native vision-language models. The full promise of NEO has remained largely untapped, hindered by scarce training data and limited computational resources, especially in knowledge-intensive and OCR-focused domains. Yet, strikingly, our NEO rivals state-of-the-art VLMs despite these severe constraints. We envision subsequent directions of NEO for the native VLM community as follows:

**Contextual relevance to recent advancements.** Recent models such as Qwen3VL highlight concepts that resonate with our design choices, including dense linking of visual-language features, relative positional encodings, and architectural details like patch embedding and bias. In particular, the DeepStack approach underscores the importance of establishing strong pixel-word associations from the earliest stages, reinforcing the significance of densely integrated visual-language representations.

**Maximizing the potential via large investment.** It is in great demand for continuously investing substantial resources, especially during the pre-training stage, to fully unlock NEO's performance and approach the upper bound of the native model. At the same time, selectively open-sourcing key components during intermediate development can reduce follow-up training costs for future researchers and attract more research to native visual-language models. Moreover, the fundamental models from this work provide a valuable baseline for advancing reinforcement learning research.

**Explorations of full-spectrum model capacities.** Expanding the full model sizes remains a critical factor in advancing various real-world applications. Even with limited resources, NEO-2.2B closely matches those of modular visual-language models with equivalent capacity, suggesting that the design philosophy of models in the 0.6 to 8 billion parameter range has matured. Such architectures not only achieve high performance but also facilitate the deployment of lightweight models at the edge, which is crucial for scenarios with limited computational resources or strict real-time requirements.

**Upgrading architectures and applications.** To date, our work has focused on dense models for image-text understanding, while a sparse divide-and-conquer architecture is simultaneously under active development. Notably, we regard NEO not merely as an autoregressive VLM but as a new paradigm for visual-language intelligence. Its principle is to leverage end-to-end training within a unified architecture, eliminating manually imposed biases and scaling-up complexities by allowing data and models to dictate the learning process. Besides, our efforts are designed not merely to improve performance but to establish a definitive baseline for visual-language generation, long video understanding, and embodied AI. Crucially, NEO's architecture systematically integrates the demands of video generation and related tasks, including attention mechanisms and rotary positional encodings, from the ground up. Although currently focused on text and images, NEO is poised to push the boundaries of what is possible across a wide spectrum of application scenarios and input modalities.

