# OpenReview forum: "From Pixels to Words -- Towards Native Vision-Language Primitives at Scale"
_ICLR.cc/2026/Conference — ICLR 2026 Poster_

### Official Review · Reviewer_EYZR · 2025-10-24

**Soundness:** 2
**Presentation:** 2
**Contribution:** 2
**Rating:** 4
**Confidence:** 4

**Summary:**

This work proposes a novel monolithic large vision-language model, called NEO, supported by an improved rotary positional embeddings (RoPE) mechanism and multi-stage training. NEO is also backed by several existing techniques, such as hybrid attention masking, and shared FFN, attention and norm layers for vision and language modalities. With evaluations reported on several well-established benchmarks, the work aims to demonstrate the performance of the proposed method.

**Strengths:**

The primary strengths of the work could be listed as the following:

- The particular idea of leveraging an improved RoPE mechanism (called "Native RoPE" in the work) in the context of monolithic vision-language models is novel.
- The work includes a very thorough literature review with more than sufficient citations to contemporary works, even those released within the same year, which is much appreciated.
- Evaluation is performed on a decent number of benchmarks, and there are a number of ablations on the positional encoding strategy, which is also a great to have.

Relatively minor strengths of the work could be listed as the following:

- Overall structure and flow of thought presented in the work is decent.
- The figures are of very high quality and are visually appealing, though a bit crowded (see the minor weaknesses below).

**Weaknesses:**

The primary weaknesses of the work could be listed as the following:

**W1: Architectural and Training-time Adjustments Similarities with Existing Works:** The work borrows heavily from two existing works, EVE [A] and EVEv2 [B] in both of its architectural and training-time adjustments. In particular, sharing the norm layers, attention blocks and FFN blocks have been explored in [A], patch embedding and word embedding layers are nearly identical to [B], and the _native multi-modal attention_ is the standard practice in large VLMs [C, D]. Furthermore, the overall training strategy is nearly identical to [B], with the Stage 1 pretraining corresponding to [B]'s Stage 1 & 2.1, Stage 2 mid-training corresponding to [B]'s Stage 2.2. and Stage 3s matching. Normally, having these similarities would not be a major weakness if it was not for the narrative of the work, which, in its current form, appears to present these as novelties of the proposed framework.

**W2: Ambiguities in the Narrative:** There are several ambiguities in the narrative of the work. Most importantly, due to the aforementioned W1 the exact contributions of the work are not very clear to read from the text, as the current narrative renames a few well-established practices in the field in a rather ad-hoc manner. To exemplify, these include renaming the standard hybrid attention masking in the literature to "Native Multi-modal Attention" or renaming the monolithic blocks of [A, B] to "Native VLM primite" while the only architectural difference from [A, B]'s blocks is the improved RoPE mechanism and the added Q and K parameters that go along with it. Finally, I found it a bit hard to grasp the exact changes introduced over existing works in Section 3.1 in general, with several terms like "Pre-Buffer" not being well-defined.

**W3: Fairness of Evaluations:** The work indeed includes a good number of benchmarks and a good number of ablations trying out different RoPE variants within the same framework. However, one critical thing that is lacking is a fair comparison between [A, B] and this work. Notably, NEO utilizes a much better LLM compared to the baselines considered (Qwen 3 versus older Qwen 2.5/Vicuna variants) in Table 1 and it was also trained with much more data than many of them. Given NEO does not add much beyond the improved RoPE mechanism over [A, B] architecturally, a more fairer comparison would demand them trained with similar budgets or at least with similar performing LLMs.

Relatively minor weaknesses of the work could be listed as the following:

- In many parts of the text the usage of \citep and \citet commands were used incorrectly. This hinders the readability of the text for the broader audience and fixing them would greatly improve the reading experience.

- Some figures are very crowded with many details, creating potential confusions in grasping their main message. To exemplify, Figure 1 includes many different details regarding the full LVLM pipeline. Although its quality is very high and I can clearly see that much effort went into constructing it, I believe that pruning it greatly would make it easier for the reader to grasp its main message.

*Finally , although I am leaning towards rejection for the work in its current form, I would like to encourage the authors to clarify any potential misunderstandings I might have had.*

---
[A] Diao, H., Cui, Y., Li, X., Wang, Y., Lu, H., & Wang, X. (2024). Unveiling encoder-free vision-language models. Advances in Neural Information Processing Systems, 37, 52545-52567.

[B] Diao, H., Li, X., Cui, Y., Wang, Y., Deng, H., Pan, T., ... & Wang, X. (2025). Evev2: Improved baselines for encoder-free vision-language models. arXiv preprint arXiv:2502.06788.

[C] Beyer, L., Steiner, A., Pinto, A. S., Kolesnikov, A., Wang, X., Salz, D., ... & Zhai, X. (2024). Paligemma: A versatile 3b vlm for transfer. arXiv preprint arXiv:2407.07726.

[D] Chen, Z., Wu, J., Wang, W., Su, W., Chen, G., Xing, S., ... & Dai, J. (2024). Internvl: Scaling up vision foundation models and aligning for generic visual-linguistic tasks. In Proceedings of the IEEE/CVF conference on computer vision and pattern recognition (pp. 24185-24198).

**Questions:**

- Given the similarities in performance with Video-RoPE [E] and the proposed RoPE mechanism in this work (Table 3), how well the proposed RoPE mechanism compare against it under smaller training budgets, such as those utilized for [A, B]?

- Can you comment on the fairness of evaluations raised from above? How do you think the differences in training and architectural settings could be effecting the evaluation results and how do you think you could address these?

---
[A] Diao, H., Cui, Y., Li, X., Wang, Y., Lu, H., & Wang, X. (2024). Unveiling encoder-free vision-language models. Advances in Neural Information Processing Systems, 37, 52545-52567.

[B] Diao, H., Li, X., Cui, Y., Wang, Y., Deng, H., Pan, T., ... & Wang, X. (2025). Evev2: Improved baselines for encoder-free vision-language models. arXiv preprint arXiv:2502.06788.

[E] Wei, X., Liu, X., Zang, Y., Dong, X., Zhang, P., Cao, Y., ... & Lin, D. (2025). VideoRoPE: What Makes for Good Video Rotary Position Embedding?. arXiv preprint arXiv:2502.05173.

---

> ### Author Response · Authors · 2025-11-21
> **Response to Reviewer EYZR (Part1)**
>
> We are grateful for the reviewer’s careful assessment and valuable suggestions. We respond to the raised concerns below.
>
> ---
>
> > **Q1: Architectural and Training-time Adjustments Similarities.**
>
> **A1:** We thank the reviewer for the detailed comparison between NEO and other native VLMs. We fully agree that some architectural elements and multi-stage training pipelines, are commonly used across recent VLMs. NEO does not claim any of these widely adopted components as contributions.
>
> - **(1) Architectural Parts**: Shared norm, attention, and FFN are standard across almost all "Dense" native methods in Tables 1–2. Patch embedding layers are also nearly identical among existing native VLMs, and Mixed attention is a commonly used practice and is not part of our claimed contributions.
> **What distinguishes NEO is the unified primitive with native RoPE and its corresponding interaction pattern by decouling spatial and temporal dimensions.** This design natively encodes, aligns, and reasons across modalities from an intrinsically multimodal perspective. Note that prior native VLMs largely follow **LLM's RoPE or causal-only design push the LLM to learn visual perception from scratch**. In contrast, NEO integrates **the strengths of both vision encoders and LLMs** into a single native multimodal primitive rather than **forcily attaching visual function onto an original LLM backbone**.
>
> - **(2) Training Parts**: Three-stage procedures (PT -> MT -> SFT) are also common across almost all modular and native VLMs in Tab.1-2. Besides, LLM-centric pre-alignment is also widely-used in existing native VLMs, e.g., EVE series, Mono-InternVL series, BREEN, VoRA, SAIL, etc..
> **We do not claim this strategy as a contribution.** NEO simply demonstrates that fully end-to-end training VLMs is also feasible for learning visual perception from scratch, requiring no visual encoder distillation (EVE, HoVLE, BREEN, OneCAT, and VoRA), or on separate pre-training stages that modular VLMs often do.
>
> **We further polish the introduction (Line 89-91), related works (Line 158-161), and Section 3.1 (Line 235-237)** to highlight our core contribution clearer and to avoid any misunderstanding.
>
>
> ---
>
> > **Q2: Ambiguities in the Narrative.**
>
> **A2:** Thanks for your meaningful advice, and we further improve Section 3.1 (Line 199-208, 235-237, 244-259) and Fig.3 to aviod ambiguities in the narrative:
>
> - **(1) Monolithic Blocks**: As discussed in Q1, NEO's contribution lies in **its unified primitive with native RoPE and multi-dimension interaction pattern. This design enables NEO to encode, align, and reason across modalities from an intrinsically multimodal perspective.** Tab.2 shows that this unified primitive yields substantial performance gains of 4.9% over prior native VLM blocks that largely inherit the RoPE and attention mechanisms of LLMs.
>
> - **(2) Pre-Buffer Parts**: It functions as an encoding layer similar to the visual encoders before LLMs, but it is shared across both vision and text. **Its role is to map visual and linguistic inputs into a unified representation space and to mitigate the disturbance that arises when raw patch and word embeddings are fed directly into the LLM.** To clarify this, we have added more formalism and detailed explanations of its mechanism and initialization in Lines 244–259 and Fig. 3.
>
>
>
> ---
>
> > **Q3: Comparison between EVEv1 and EVEv2.**
>
> **A3:** As noted in Tab.1-2, existing native VLMs differ substantially in training pipelines, data volume/quality, and base LLM choice, making full from-scratch retraining of all models prohibitively expensive.
> Our ablations isolate the architectural factors (native RoPE/attention in Tab.3, pre-Buffer in Fig.5) that distinguish NEO with existing native works.
>
> To further clarify it, we retrain the representative EVEv1.0 (dense), EVEv1.5 (Mixture-of-Experts), and EVEv2.0 (Divide-and-Conquer) based on Qwen3-1.7B under the unified training setup in Tab.3.
> The new experiment in Fig.7 shows that **EVEv1.0, EVEv1.5, and EVEv2.0 achieves 33.4%, 40.2%, and 41.5% vs. 44.0% of NEO.** This confirms that the gains arise from **our pre-buffer, native RoPE design, and multi-dimensional interaction patterns** rather than from the newer LLMs and data volume/quality **alone**.  We hope these controlled comparisons and additional results can address the concern.

---

> ### Author Response · Authors · 2025-11-21
> **Response to Reviewer EYZR (Part2)**
>
> > **Q4: Citation and Figure contents.**
>
> **A4:** Thanks for your constructive suggestions. We have revised the citation and simplified the figures to make the content clearer and more readable.
>
> ---
>
> > **Q5: Comparisons with Video-RoPE using the smaller budgets, such as those utilized for [A, B].**
>
> **A5:** Tab.3 already evaluates the RoPE variants under a small data budget (**20M PT and 2M SFT data**), which is smaller than [A,B] using **33–77M PT and 1.8–7M SFT data**.
> Following [A, B], we further enlarge the training corpus to include **20M PT, 15M MT, and 2M SFT data**, resulting in a comparable budget. Notably, the performance gains slightly increase **from 0.8% to 1.1% on average across 10 benchmarks** in Tab.3, demonstrating that the performance benefits of native RoPE design are relatively consistent.
>
> ---
>
> > **Q6: The fairness of evaluations raised from above.**
>
> **A6:** Please refer to our response to Q3 for detailed clarification.

---

> > ### Comment · Reviewer_EYZR · 2025-11-25
> >
> > I thank the authors for the efforts they put into their rebuttal.
> >
> > For the first weakness, I initially raised the issue of similarities in narrative to existing works, particularly to EVE and EVEv2. As a response, the authors have updated several parts of the text to highlight their contributions more clearly, particularly in the introduction and the related work sections. After going through the updates, I believe that the narrative of the work is now clearer and my concerns on this end are mostly resolved.
> >
> > As a second weakness, I initially found several parts of the work to be hard to go through, especially regarding the exact contributions of the work and several parts of the methodology in Section 3. Similarly to the first weakness, the authors have updated their main text to make these parts easier to grasp and more crisp. Similarly, my concerns on this end are also mostly resolved.
> >
> > Finally, as the last major weakness, I raised the issue of unfair comparisons against older and similar variants due to older baselines' usage of older LLMs and NEO's usage of the newest LLMs, most notably against EVE and EVEv2. As a response, the authors have benchmarked the average performance of these baselines and their NEO under a fair setting of using the same LLM and quantitatively demonstrated the improvements of NEO.
> >
> > As all of my major concerns have been resolved, I am increasing my score to a clear acceptance. I am not recommending an even higher score yet as I believe that the methodological novelty of the work to be limited to a novel RoPE variant.

---

> > > ### Author Response · Authors · 2025-11-26
> > > **Response to Reviewer EYZR**
> > >
> > > Thanks for your thoughtful and constructive feedback. We are glad that the revisions have addressed your concerns regarding the narrative clarity, methodology, and fairness of comparisons. We appreciate your positive assessment and the improved score. Your comments have helped us strengthen the paper, and we are grateful for your support.

---

### Official Review · Reviewer_JqhJ · 2025-10-27

**Soundness:** 3
**Presentation:** 4
**Contribution:** 4
**Rating:** 8
**Confidence:** 3

**Summary:**

The paper proposes a novel approach to train autoregressive, monolithic vision-language models that omit domain-specific vision encoders in favor of light-weight encoders and a native multi-modal training resulting in the *NEO* models.  The model consists of a small two-layer convolution encoder, a multimodal pre-buffer, and a pretrained LLM (Qwen3). The attention and position encoding procedure is optimized for multi-modality. The model training comprises of 3 stages with 390M image-text examples. The resulting 2.2B/9B models are thoroughly benchmarked against prior modular and monolithic VLMs, and outperform all prior models in the latter category.

**Strengths:**

- The paper proposes improvements in native, monolithic multimodal LLM training by introducing a pre-Buffer for better alignment (trained separately in the first stage), native rotary position embeddings with modality-specific base frequencies, native multimodal attention (causal for text, full-bidirectional for vision) with decoupled H, W, T processing.
- Using 390M image-text examples, NEO reaches a high performance and outperforms all previous native VLMs.
- NEO is built on top of a modern LLM (Qwen3) and supports flexible resolution
- The authors provide NEO-2B and 9B intermediate and final checkpoints
- The models are thoroughly evaluated against other models, including modular and native models in 2B and 8/9B categories
- Some design choices (number of layers in the pre-Buffer and attention/embedding methods) are ablated

**Weaknesses:**

- I keep wondering why NEO-9B uses a 50% smaller pre-Buffer than 2.2B. The paper mentions "mainly due to the good trade-off between performance and efficiency." (L431) but does not provide evidence for that. I am not convinced that the results in Fig. 5 extrapolate to a larger post-LLM.
- Some systematic ablations of design choices are often missing. The number of layers in the pre-Buffer and attention/embedding methods are ablated but nothing else. This left wondering which design choices in NEO actually impact performance: e.g., is it the data (quality/quantity)? The stages? The more modern LLM (related work uses older LLMs)? Or, is it actually the proposed combination of design choices. I understand that providing such controlled ablation experiments might not be economically feasible, but they obfuscate the contribution nonetheless.
- NEO still significantly lags behind modular VLMs, even older ones like Qwen2-VL (e.g., 16% on InfoVQA). Given the relative improvements in its category and the lower amount of data this is not a big issue itself, however the phrasing in "Comparison with Modular VLMs." (L357ff) is a bit overselling.
- The scaling improvements between 2B and 9B seem modest compared to modular VLMs, "casting shadows" over the scalability of NEO.
- Some parts of the paper feel LLM-generated by overusing (sometimes nonsensical) synonyms, making it hard to follow the paper. I would encourage the authors to manually revise the paper.
- Fig. 1/3 are densely packed and hard to comprehend.

**Questions:**

- Are there any insights why the performance on HallusionBench and "knowledge-heavy" tasks suffers? Fundamentally, this does not seem like a multi-modal problem to me.
- Please review the LLM written parts for clarity.
- Please consider using \citep to improve legibility

---

> ### Author Response · Authors · 2025-11-21
> **Response to Reviewer JqhJ (Part1)**
>
> We sincerely appreciate the reviewer’s thoughtful and constructive feedback. Our detailed responses are provided below.
>
> ---
>
> > **Q1: Why NEO-9B uses a 50% smaller pre-Buffer.**
>
> **A1:** Good question!
> **(1)** The size of the Pre-Buffer is inherently coupled to the base LLM, as its native primitives are directly on top of pre-trained LLM layers.
> **Here integrating a 12-layer Pre-Buffer into Qwen3-8B introduces roughly 2.4B extra parameters**. Such an extremely heavy configuration makes it highly challenging to fully training Pre-Buffers with 8–12 layers, considering our current limited pre-training data scale.
>
> **(2)** Notably, recent work [a] systematically investigates a **joint scaling law between LLMs and VEs, validating that the optimal visual encoder size (1.2B parameters) scales proportionally with 7–8B LLMs.** Inspired by this, we select a 6-layer Pre-Buffer as a practical compromise between performance and efficiency under the available data and compute budget.
>
> **(3)** We have incorporated these considerations into **Section 3.1 (Line 244-248) and ablation study (Line 427-431)**.
>
> [a] Changyao Tian, et al. NaViL: Rethinking Scaling Properties of Native Multimodal Large Language Models under Data Constraints (NeurIPS 2025)
>
>
> ---
>
> > **Q2: Which design choices in NEO actually impact performance?**
>
> **A2:** Thanks for your valuable advice. As shown in Tables 1–2, existing native VLMs differ substantially in training pipelines, data scale and quality, and base LLMs. Testing various controlled ablation experiments would be prohibitively expensive, and **we appreciate the reviewer’s understanding of these constraints.**
>
> To further clarify our architectural contributions, we retrain the representative EVEv1.0 (dense), EVEv1.5 (Mixture-of-Experts), and EVEv2.0 (Divide-and-Conquer) based on Qwen3-1.7B under the unified training setup in Tab.3.
> The new experiment in Fig.7 shows that **EVEv1.0, EVEv1.5, and EVEv2.0 achieves 33.4%, 40.2%, and 41.5% vs. 44.0% of NEO**. This confirms that the gains arise from **our pre-buffer, native RoPE design and interaction patterns**, rather than from the newer LLMs and larger and high-quality data **alone**.  We hope these controlled comparisons and additional results can address the concern.
>
>
> ---
>
> > **Q3: Weak performance on on InfoVQA, and the phrasing in "Main Results".**
>
> **A3:** We sincerely appreciate the reviewer’s understanding about the relatively limited amount of data used in our setting. We agree that although NEO has largely narrowed the gap with top modular models (e.g., InternVL3), performance on several datasets remains limited.
>
> (1) **We have polished the Abstract (Line 23-24), Introduction (Line 104-107), and Main Results (Line 361-368) accordingly, using clearer descriptions** such as “narrow the gap with modular counterparts” and “approaches modular models on diverse benchmarks.”.
>
> (2) **To isolate architectural impact from data effects, we have trained two modular counterparts (“Encoder-Based” in Tab.1–2)** by combining Qwen3 and InternViT-300M under identical mid-training (MT) and supervised finetuning (SFT) data conditions.
>
> |Model| PT/MT/SFT-Data |7 VL tasks|6 VQA tasks|InfoVQA|
> |--|--|--|--|--|
> |(2B) NEO|345M/40M/4M|61.89|77.58|63.2|
> |(2B) **Encoder-Based**|>6B/40M/4M|59.7|78.1|65.9|
> |(2B) **NEO+**|345M/58M/4M|63.60|78.65|64.8|
> |(2B) InternVL3|>6B/100M/22M|65.8|78.97|66.1|
> |(9B) NEO|345M/40M/4M|66.26|77.90|60.9|
> |(9B) **Encoder-Based**|>6B/40M/4M|68.1|82.7|75.0|
> |(9B) **NEO+**|345M/58M/4M|68.81|80.54|65.3|
> |(9B) InternVL3|>6B/100M/22M|73.57|84.92|76.8|
>
> - *Under matched MT and SFT data, NEO-2B rivals its modular counterparts on most benchmarks, including on MMMU and MMVet, despite using fewer pre-training samples.* Notably, NEO-9B shows the similar performance trend.
>
> - *With the larger 58M MT dataset, NEO shows extra performance gains on general VL and VQA benchmarks, further narrowing the gap.* Besides, NEO-9B benefits more than NEO-2B, suggesting that NEO remains undertrained given current data limitations.
>
> Together, these findings show that **the remaining gap to top-tier modular models is driven primarily by data scale and quality.** In future work, we will continue expanding the data corpus to further unlock NEO’s potential.

---

> ### Author Response · Authors · 2025-11-21
> **Response to Reviewer JqhJ (Part2)**
>
> > **Q4: The scalability of NEO.**
>
> **A4:** Good questions! As we mentioned in Q1 and Q3, current 390M–scale training data used for NEO-2B is not sufficient, and this limitation becomes even more pronounced for NEO-9B, since they must learn visual perception entirely from scratch. **With additional training data, NEO-9B demonstrates more advantages over NEO-2B, indicating substantial potential for continued data scaling.**
>
> |Model| PT/MT/SFT-Data |7 VL tasks|6 VQA tasks|InfoVQA|
> |--|--|--|--|--|
> |(2B) NEO|345M/40M/4M|61.89|77.58|63.2|
> |(2B) **Encoder-Based**|>6B/40M/4M|59.7|78.1|65.9|
> |(2B) **NEO+**|345M/58M/4M|63.60|78.65|64.8|
> |(2B) InternVL3|>6B/100M/22M|65.8|78.97|66.1|
> |(9B) NEO|345M/40M/4M|66.26|77.90|60.9|
> |(9B) **Encoder-Based**|>6B/40M/4M|68.1|82.7|75.0|
> |(9B) **NEO+**|345M/58M/4M|68.81|80.54|65.3|
> |(9B) InternVL3|>6B/100M/22M|73.57|84.92|76.8|
>
> **We sincerely appreciate the reviewer’s understanding of our data limitations.** We view NEO as a starting point and look forward to future work with greater resources and larger-scale data to further uncover the potential of native VLMs, as discussed in the Limitations and Discussion section.
>
>
> ---
>
> > **Q5: Writing, citation, and Fig.1/3.**
>
> **A5:** Thank you for your careful reading and constructive suggestions. We have revised the relevant text and simplified the figures to improve clarity and legibility.
>
>
> ---
>
> > **Q6: Any insights towards the performance on HallusionBench and "knowledge-heavy" tasks.**
>
> **A6:** We agree that **this issue is not fundamentally a multi-modal problem**.
> By comparing NEO-2B/8B with their modular counterparts in the new Table 1, we observe that NEO performs on par with both benchmarks. This suggests that the performance on HallusionBench and knowledge-heavy tasks is likely dominated by data-related and LLM-intrinsic factors over multimodal components.

---

### Official Review · Reviewer_QS4o · 2025-11-01

**Soundness:** 2
**Presentation:** 2
**Contribution:** 3
**Rating:** 6
**Confidence:** 3

**Summary:**

The authors present Neo, a family of native Vision-Language Models built on top of Qwen3-1.7B and Qwen3-8B. The key elements of the Neo architecture are (1) an attention block that decouples H, W and T dimensions for Query and Key computation; (2) RoPE position embeddings that use separate frequencies for H, W and T dimensions; (3) bidirectional attention for images; (4) added transformer layers ("Pre-Buffer") to project vision and text embeddings to the same embedding space. The authors train Neo models on 390M image-text examples in a process containing three stages and compare against prior modular VLMs using pretrained vision encoders and native VLMs. The trained models obtain competitive performance on various datasets against modular VLMs, while outperforming prior work on native VLMs.

**Strengths:**

- Each architectural component is motivated with the incorporation of inductive biases that make the processing of images in native VLMs more similar to modular VLMs, which I found to be intuitive.
- The decoupling of H, W and T dimensions in both the Query and Key computation as well for RoPE computation is novel to the best of my knowledge.
- The training process does not assume access to any pretrained vision encoders.
- The various ablation studies in Section 4.3 are very valuable in arguing for the importance of various design choices in the attention blocks, particularly of the suggested adjustment to RoPE. This is especially the case when all native VLM models compared against make use of different datasets or pre-trained LLMs.

**Weaknesses:**

While I believe the contribution is valid, in large part due to the ablation studies, there are various flaws to the paper:
- Firstly, there are clear errors in the related works section. We do not know the underlying architecture for multimodal GPT models and it is therefore incorrect to claim that they are modular or native. It is also important to note that GPT-4o, being able to both condition on and generate images, is more similar to a model like Chamelon (included in the native model section) than standard modular model counterparts. It is likewise an issue to make claims regarding Claude or Gemini.
- I would argue that saying Neo "rivals top-tier modular counterparts" in the abstract and introduction is overclaiming. For general vision benchmarks, Neo approaches modular model performance, but nonetheless falls short in each case. This is particularly an issue for MMMU and MMVet, where 8B model performance is at least ~20% poorer. This and the bullet point above are the rationale for the soundness score.
- Likewise, any comparison made to prior native VLMs has the confounder of Neo making use of the newer and stronger Qwen3 backbones. This is mitigated by the ablation studies, which are much welcome.
- I found the writing to be unclear, particularly for sections dealing with the "Pre-Buffer" and "Post-LLM." Some added formalism about what computation each of these components perform and which components are initialized from the Qwen3 backbone and which are not would make understanding easier. Figure 2 sadly is unclear as both the Pre-Buffer and Post-LLM components make use of the same primitives but just differ in color.

**Questions:**

- A majority of the citations should be changed to parentheticals, rather than in-text citations with \citet.
- As a follow-up to my point regarding the Pre-Buffer, I also struggled at understanding what exactly was done in the "Comparison between Pre-Buffer and Vision Encoders" section. Would it be correct to say that training was repeated here with InternViT/CLIP/SigLIP used in place of the Pre-Buffer?

---

> ### Author Response · Authors · 2025-11-21
> **Response to Reviewer QS4o (Part1)**
>
> We sincerely thank the reviewer for the supportive and insightful comments. We address the valuable points in detail below.
>
> ---
>
> > **Q1. Explain some claims in the related works section.**
>
> **A1:** Thank you for the constructive feedback. We mainly follow the categorizations in prior native multimodal works (e.g., Mono-InternVL series [a,b], EVE series [c,d]).
> **To avoid ambiguity, we have revised the related-work section in Line 131-133** by replacing those models with ones whose architectural details are clearly documented, such as **Seed-VL and GLM-V**.
>
> [a] Gen Luo, et al. Mono-InternVL: Pushing the Boundaries of Monolithic Multimodal Large Language Models with Endogenous Visual Pre-training. CVPR2025
>
> [b] Gen Luo, et al. Mono-InternVL-1.5: Towards Cheaper and Faster Monolithic Multimodal Large Language Models. Arxiv2507.12566
>
> [c] Haiwen Diao, et al. Unveiling Encoder-Free Vision-Language Models. NeurIPS2024
>
>
> [d] Haiwen Diao, et al. EVEv2: Improved Baselines for Encoder-Free Vision-Language Models. ICCV2025
>
>
> ---
>
> > **Q2. Descriptions of NEO in the abstract and introduction.**
>
> **A2:** We agree that although NEO has largely narrowed the gap with top modular models (e.g., InternVL3), performance on serveral datasets remains limited.
>
> (1) Thanks for the valuable suggestions. **We have polished the Abstract (Line 23-24), Introduction (Line 104-107), and Main Results (Line 361-368) accordingly, using clearer descriptions** such as “narrow the gap with modular counterparts” and “approaches modular models on diverse benchmarks.”.
>
> (2) **To isolate architectural impact from data effects, we have trained two modular counterparts (“Encoder-Based” in Tab.1–2)** by combining Qwen3 and InternViT-300M under identical mid-training (MT) and supervised finetuning (SFT) data conditions.
>
> |Model| PT/MT/SFT-Data |7 VL tasks|6 VQA tasks|
> |--|--|--|--|
> |(2B) NEO|345M/40M/4M|61.89|77.58|
> |(2B) **Encoder-Based**|>6B/40M/4M|59.7|78.1|
> |(2B) **NEO+**|345M/58M/4M|63.60|78.65|
> |(2B) InternVL3|>6B/100M/22M|65.8|78.97|
> |(9B) NEO|345M/40M/4M|66.26|77.90|
> |(9B) **Encoder-Based**|>6B/40M/4M|68.1|82.7|
> |(9B) **NEO+**|345M/58M/4M|68.81|80.54|
> |(9B) InternVL3|>6B/100M/22M|73.57|84.92|
>
> - *Under matched MT and SFT data, NEO-2B rivals its modular counterparts on most benchmarks, including on MMMU and MMVet, despite using fewer pre-training samples.* Notably, NEO-9B shows the similar performance trend.
>
> - *With the larger 58M MT dataset, NEO shows extra performance gains on general VL and VQA benchmarks, further narrowing the gap.* Besides, NEO-9B benefits more than NEO-2B, suggesting that NEO remains undertrained given current data limitations.
>
> Together, these findings show that **the remaining gap to top-tier modular models is driven primarily by data scale and quality.** In future work, we will continue expanding the data corpus to further unlock NEO’s potential.
>
>
> ---
>
> > **Q3: Comparisons with prior native VLMs using stronger Qwen3.**
>
> **A3:** As noted in Tab.1-2, existing native VLMs differ substantially in training pipelines, data volume/quality, and base LLM choice, making full from-scratch retraining of all models prohibitively expensive.
> **This is precisely why our ablations isolate the architectural factors (native RoPE/attention in Tab.3, pre-Buffer in Fig.5) that distinguish NEO, and we are grateful that the reviewer has noted them.**
>
> To further clarify it, we retrain the representative EVEv1.0 (dense), EVEv1.5 (Mixture-of-Experts), and EVEv2.0 (Divide-and-Conquer) based on Qwen3-1.7B under the unified training setup in Tab.3.
> The new experiment in Fig.7 shows that **EVEv1.0, EVEv1.5, and EVEv2.0 achieves 33.4%, 40.2%, and 41.5% vs. 44.0% of NEO**. This confirms that the gains arise from **our pre-buffer, native RoPE design, and multi-dimensional interaction patterns** rather than from the newer LLM backbone **alone**.  We hope these controlled comparisons and additional results can address the concern.

---

> ### Author Response · Authors · 2025-11-21
> **Response to Reviewer QS4o (Part2)**
>
> > **Q4: Added formalism about Pre-Buffer and Post-LLM.**
>
> **A4:** Thanks for your valuable advice. We have polished the related manuscript in Section3.1 (Line 224-259), and Fig.3(4) for detailed pre-Buffer and post-LLM illustration.
>
> (1) Functional Descriptions. Both the Pre-Buffer and Post-LLM modules, built from the same native primitives (new Eq.(3)), sequentially process visual patch embeddings and word embeddings within a single monolithic backbone.
>
> (2) Initialization details. The detailed implementations are shown in the improved Fig.3(4). For clarity:
>
> - **Pre-Buffer:** **fully randomly initialized**.
> - **Post-LLM:** components including RMSNorm, FFN layers, \(Q/K/V/QK-Norm\) in temporal dimension are **initialized from the pre-trained Qwen3**.
>   Spatial dimensions follow a principled initialization strategy:
>   - \(Q_H\) and \(Q_W\) are initialized by copying **pretrained \(Q_T\)**.
>   - \(K_H\) and \(K_W\) are **zero-initialized**.
>   - QK-Norm for spatial axes uses **\($\beta$ = 0\) and \($\gamma$ = 1\)**.
>
> ---
>
> > **Q5: Citation format.**
>
> **A5:** Thank you for pointing this out. We have updated the citation style to *citep* to improve legibility.
>
> ---
>
> > **Q6: Comparison between Pre-Buffer and Vision Encoders in Fig.6.**
>
> **A6:** Yes. We extract Pre-Buffers from three stages, denoted PB1–3. For a fair comparison, we pair PB1–3, InternViT, CLIP, and SigLIP with identical post-LLM backbones initialized from Qwen3-1.7B, and retrain all resulting VLMs using the same 20M pre-training and 2M SFT datasets.
> **Notably, PB1–3 process images and text jointly, whereas InternViT, CLIP, and SigLIP operate solely on visual inputs, following the design of encoder-based VLMs.**
> Despite being pre-trained on substantially larger image–text corpora, these vision encoders offer only limited gains over the Pre-Buffer variants. This underscores the strength and efficiency of Pre-Buffer as an initialization strategy for future native VLM research.
>
> **We have refined Fig.6 and corresponding discussions (Line 467-475) to present the findings more clearly.**

---

### Author Response · Authors · 2025-11-21
**General Response**

We thank all reviewers for their careful and constructive feedback. Reviewers consistently recognize our **novel approach** (QS4o, JqhJ, EYZR), **strong performance** (QS4o, JqhJ), **comprehensive comparisons** with existing methods (QS4o, JqhJ, EYZR), **valuable ablation studies** (QS4o, JqhJ, EYZR), and **concise training strategy** (QS4o).

We have polished the manuscript and improved the illustrations. We hope that these revisions adequately resolve the main issues and kindly invite the reviewers to re-evaluate the paper based on our responses and the updated version.

The main revisions to the paper are summarized as follows:

- **Clear comparisons with prior native VLMs:** We have added detailed comparisons between NEO and its native counterparts under the same data budgets and LLM backbone (Fig. 7; Lines 476–482).

- **Claims regarding performance against modular models:** We have revised the abstract (Lines 23–24), introduction (Lines 104–107), and main results (Lines 360–368) to clarify and accurately qualify these claims.

- **Clearer highlight of contributions:** We have further polished the statements of key contributions in the introduction (Lines 89–91), related work (Lines 158–161), and methodology (Lines 235–237).

- **Formalization of several concepts:** We have revised and clarified the definitions of primitives (Lines 199–208) and the pre-Buffer and post-LLM components (Lines 244–259).

- **Category in related work:** We have refined the discussion to better distinguish private VLMs from open-source ones in the related work section (Lines 130–133).

- **More details between pre-Buffer and vision encoders:** Additional implementation details have been added in the ablation study section (Lines 467–475).

- **Hyper-parameters of the pre-Buffer:** We have provided further explanations in both the methodology (Lines 246–248) and ablation study (Lines 427–431).

- **Citation style, writing clarity, and figure simplification:** We have standardized citation formatting (\citep), improved related descriptions, and simplified figures to enhance readability.

---

### Author Response · Authors · 2025-12-03
**Rebuttal Summary and Consensus**

Dear Reviewers, AC, SAC, and PC,

We are deeply grateful for your thoughtful time and consideration. It is encouraging that ***all reviewers*** had recognized the novelty and contributions of our NEO and ***recommended acceptance*** up to ***Nov 25***. The post-rebuttal consensus was reflected in the rating score of ***688*** (original 684) prior to the OpenReview bug.

---

We thank all reviewers for their positive feedback, including recognition of:

- **Novel** approach (QS4o, JqhJ, EYZR) and **intuitive** motivation (QS4o).

- **Valuable** ablation studies that clearly evaluate key components (QS4o, JqhJ, EYZR).

- **Thorough** literature, benchmarks, and comparisons showing **high performance** against leading modular VLMs and over prior native VLMs (QS4o, JqhJ, EYZR).

- The training process requires **no pretrained vision encoders** (QS4o).

- **Decent** structure and thought flow, **high-quality** and **visually appealing** figures (EYZR).

---

We carefully addressed every concern and substantially strengthened the paper as follows. Notably, after we resolved all of Reviewer EYZR’s issues, the reviewer raised the score ***from 4 to 8 at 22:48 on 25 Nov 2025 (AOE)***.

- **Clear comparisons with prior native VLMs or Some ablations of design choices (QS4o, JqhJ, EYZR):** We have further discussed the ablation studies in Fig.5 and Fig.6 (also noted and acknowledged by QS4o and JqhJ).
Besides, we have added detailed comparisons between NEO and its native counterparts under the same data budgets and LLM backbone (Fig. 7; Lines 476–482) to better highlight the advantages of NEO's design choices.

- **Claims regarding performance and scaling improvements against modular models (QS4o, JqhJ):**
We have revised the abstract (Lines 23–24), introduction (Lines 104–107), and main results (Lines 360–368) to clarify and accurately qualify these claims.
Besides, we have trained two modular counterparts (“Encoder-Based” in Tab.1–2) to isolate architectural impact and identify the remaining gap due to data scale and quality. Moreover, with more datasets, NEO further narrows these gaps and benefits more at larger model scales.

- **Clearer narrative of contributions (EYZR):** We have further explained and polished the statements of key contributions in the introduction (Lines 89–91), related work (Lines 158–161), and methodology (Lines 235–237).

- **More details between pre-buffer and vision encoders (QS4o):** Additional implementation details have been added in the ablation study section (Lines 467–475).

- **Hyper-parameters of the pre-Buffer (JqhJ):** We have provided more explanations inspired by scaling-law analysis [a] and polished the descriptions in both the methodology (Lines 246–248) and ablation study (Lines 427–431).

> [a] Changyao Tian, et al. NaViL: Rethinking Scaling Properties of Native Multimodal Large Language Models under Data Constraints (NeurIPS 2025)

- **Performance on HallusionBench and "knowledge-heavy" (JqhJ):** We have updated Tab.1 to clarify this data-related and LLM-intrinsic issue.

- **Further comparisons with Video-RoPE using smaller budgets (EYZR):** We have further investigated this setting and demonstrated additional benefits.

- **Writing presentation**

  - **Category in related work (QS4o):** We have clarified the claims built upon the Mono-InternVL and EVE series and refined the descriptions in the related work section (Lines 130–133) to eliminate ambiguity.

  - **Added formalism (QS4o):** We have polished the definitions of primitives (Lines 199–208) and the pre-Buffer and post-LLM components (Lines 244–259).

  - **Citation style (QS4o, JqhJ, EYZR):** We have standardized citation formatting from (\cite) to (\citep).

  - **Writing synonyms and figure simplification (JqhJ, EYZR):** We have improved related descriptions and simplified Fig.1/3 to enhance readability.

---

We believe these updates further strengthen the paper and clarify our contributions, and we once again thank the reviewers, AC, SAC, and PC for their thoughtful assessments and professional oversight.

Best regards,

Authors of Paper #3312

---

### Meta-Review · Area_Chair_sVAW · 2026-01-06

**Summary:**

Paper was reviewed by there reviewers, receiving: 1 x marginally above the acceptance threshold, 1 x marginally below the acceptance threshold and 1 x accept, good paper ratings. All reviewers hail strong motivation for the architectural component and overall strong performance. Main concerns can be characterized as follows:

1) Issues with clarity and specificity of exposition in Related Work [QS4o] and with resect to Novelty Claims [EYZR]
2) Potential over-claiming of empirical gains [QS4o]
3) Lack of clarity in certain parts of exposition [QS4o, EYZR]
4) Fairness of comparison with other models [QS4o, JqhJ, EYZR]
5) Lacking certain ablations [JqhJ]
6) Still relative lack of performance compared to modular VLMs [JqhJ]
7) Limited scaling performance improvements when going from 2B to 9B model [JqhJ]

Authors have addressed these concerns in the rebuttal and with revisions in the paper. Specifically, [EYZR] confirms that his/her concerns have been fully addressed and mentions willingness to increase the score to "accept". Other reviewer comments have also been throughly addressed in the opinion of AC. Overall, this is an interesting paper, with valuable design insights. While the overall novelty of the work is somewhat limited, it will make a valuable contribution to the community. Therefore AC is recommending acceptance.

**Reviewer Concerns:**

1) Issues with clarity and specificity of exposition in Related Work [QS4o] and with resect to Novelty Claims [EYZR]
2) Potential over-claiming of empirical gains [QS4o]
3) Lack of clarity in certain parts of exposition [QS4o, EYZR]
4) Fairness of comparison with other models [QS4o, JqhJ, EYZR]
5) Lacking certain ablations [JqhJ]
6) Still relative lack of performance compared to modular VLMs [JqhJ]
7) Limited scaling performance improvements when going from 2B to 9B model [JqhJ]

**Reviewer Scores:**

[EYZR] explicitly mentions that he/she will increase the score to "accept", resulting in ALL accept ratings. I further believer the remaining two reviewers would likely increase their scores as well, given the content of the rebuttal (would certainly NOT decrease them).

---

### Decision · Program_Chairs · 2026-01-26

Accept (Poster)